# Surgical Treatment in Hidradenitis Suppurativa

**DOI:** 10.3390/jcm11092311

**Published:** 2022-04-21

**Authors:** Ratnakar Shukla, Priyanka Karagaiah, Anant Patil, Katherine Farnbach, Alex G. Ortega-Loayza, Thrasivoulos Tzellos, Jacek C. Szepietowski, Mario Giulini, Hadrian Schepler, Stephan Grabbe, Mohamad Goldust

**Affiliations:** 1Department of Dermatology, All India Institute of Medical Sciences Gorakhpur (AIIMS), Gorakhpur 273008, India; ratnakarshukla88@gmail.com; 2Department of Dermatology, Bangalore Medical College and Research Institute, Bangalore 560002, India; pri20111992@gmail.com; 3Department of Pharmacology, Dr. DY Patil Medical College, Navi Mumbai 400706, India; anantd1patil@gmail.com; 4Department of Dermatology, Oregon Health & Science University, Portland, OR 97239, USA; farnbach@ohsu.edu (K.F.); ortegalo@ohsu.edu (A.G.O.-L.); 5Department of Dermatology, NLSH University Hospital, 8092 Bodø, Norway; ltzellos@googlemail.com; 6European Hidradenitis Suppurativa Foundation e.V., 06847 Dessau, Germany; 7Department of Dermatology, Venereology and Allergology, Wroclaw Medical University, 50-367 Wroclaw, Poland; 8Department of Dermatology, University Medical Center Mainz, 55131 Mainz, Germany; mario.giulini@unimedizin-mainz.de (M.G.); hadrian.schepler@unimedizin-mainz.de (H.S.); stephan.grabbe@unimedizin-mainz.de (S.G.)

**Keywords:** hidradenitis suppurativa, apocrine gland, follicular occlusion

## Abstract

Hidradenitis suppurativa (HS) is a chronic, progressive inflammatory disorder of follicular occlusion with pubertal onset that presents as painful inflammatory nodules, sinus tracts, and tunnelling in apocrine-gland-rich areas, such as the axilla, groin, lower back, and buttocks. The disease course is complicated by contractures, keloids, and immobility and is often associated with a low quality of life. It is considered a disorder of follicular occlusion with secondary inflammation, though the exact cause is not known. Management can often be unsatisfactory and challenging due to the chronic nature of the disease and its adverse impact on the quality of life. A multidisciplinary approach is key to prompt optimal disease control. The early stages can be managed with medical treatment, but the advanced stages most likely require surgical intervention. Various surgical options are available, depending upon disease severity and patient preference. In this review an evidence-based outline of surgical options for the treatment of HS are discussed. Case reports, case series, cohort studies, case-control studies, and Randomized Clinical Trials (RCT)s available in medical databases regarding surgical options used in the treatment of HS were considered for the review presented in a narrative manner in this article.

## 1. Introduction

Hidradenitis suppurativa (HS), also known as acne inversa or Verneuil’s disease, is a debilitating, chronic, recurrent inflammatory disorder involving the epithelial lining of the hair follicle. It affects areas of friction that are rich in apocrine glands, such as the axilla, groin, inframammary area, and other intertriginous areas. It is clinically characterized by painful nodules of varying size, abscesses, tunnel formation, and scarring [1]. According to a systematic review, the overall prevalence of HS is 0.40, with differences based on the types of studies (0.3% in population-based studies versus 1.7% in clinical samples) [2]. HS is more common in young adults and in females compared to males (3:1–4:1) [3]. HS is characterized by recurrent flares that may be associated with foul-smelling discharge and pain, leading to a significant impairment of quality of life [4,5]. The diagnostic criteria of HS are defined by the European S1 Guideline [6]. Several scoring systems are proposed for disease severity assessment, namely, Hurley staging, the HS Physician’s Global Assessment (PGA), the modified Sartorius score (MSS), the HiSCR (hidradenitis suppurativa clinical response), and the Hidradenitis Suppurativa Severity Index (HSSI) [7,8]; however, each of these bears some limitations. Treatment of HS is distinctly challenging, as the disease involves chronic inflammation that leads to the formation of tunnels in the skin. For this reason, the majority of patients require a combination of medical therapies to reduce the inflammation and surgical treatment to address the tunnels and scarring. Medical management is not curative, but surgical intervention may have some “curative” effect; however, recurrence rates depend on the procedure and the concomitant medical management. Despite surgical interventions, the complete cure rate is very low, and relapses are quite common [7].

## 2. Perioperative Evaluation and Outcome Expectations

The clinical stratification of HS lesions is the most essential step in the evaluation of patients. The stage can be described using the Hurley staging system and inflammatory lesion count, as per the recommendations from the North American clinical management guidelines for HS [9] and the British Association of Dermatologists guidelines for the management of hidradenitis suppurativa [10]. Pain scoring using a visual analog scale (VAS) and dermatology quality of life index (DQLI) assessment prior to treatment provide a baseline reference against which to measure treatment efficacy and patient satisfaction [9,10,11]. Quality of life assessment tools such as HiSQOL [12], HSQoL-24 [13], or others [5] are specific for HS. Validated versions for the population in a specific country can be used for assessment of the quality of life. A thorough clinical history and screening for habits such as smoking and comorbidities, including cardiovascular disorders, metabolic diseases, depression, spondyloarthropathy and Crohn’s disease, are also key in achieving optimal outcomes, with an aim to provide adjuvant treatment and lifestyle modifications [11]. Pre-surgical evaluation with ultrasound is useful for improving the delimitation of surgical margins and reducing the post-surgical recurrence rate [14]. 

The choice of surgery is dependent on many factors, such as the extent and severity of disease, recurrence in previously managed sites (either medical or surgical), ease of operability at an affected site, experience of the operating surgeon, and patient preferences. Factors such as intraoperative and postoperative complications, time to wound healing, recurrence rates, and patient satisfaction in terms of reduction in pain score and DQLI should be considered before planning surgery [15]. A lack of controlled trials and defined outcome measures for their comparison makes it difficult to reach a consensus on the choice of surgery in different case scenarios, which is currently left to the operating surgeon’s best judgement. 

## 3. Types of Surgical Treatment 

There are various options available for surgical intervention, with no optimal treatment, requiring an individualized approach for each patient. The choice of surgical treatment depends upon various factors, such as the chronicity and extent of disease, affected site, presence of long-standing lesions, and patient comorbidities.

Surgical treatment in HS ranges from procedural treatments (e.g., laser) and minor surgery (e.g., incision and drainage and deroofing) to major surgery (e.g., wide local excision).

### 3.1. Incision and Drainage (I and D)

Incision and drainage can be used as a treatment modality in acute cases with tender fluctuant abscesses with pus accumulation. It offers the opportunity for rapid pain relief. However, this modality does not actively intervene in the diseased tissue; as such, the relief is temporary and is associated with a nearly 100% recurrence rate [15,16,17]. 

After wide circumferential local anesthesia, an incision is performed and digital pressure is applied to express the purulent contents. Saline washes can be administered to extrude the remaining contents. Packing is not required following the drainage of HS lesions, but it can be considered in cases when an infectious abscess is possible [18]. Currently, only a few case reports are available for incision and drainage procedures, and it lacks substantial evidence supporting its use in HS. As deroofing can be performed with the same equipment and requires approximately the same amount of time, experts recommend deroofing over incision and drainage.

### 3.2. Deroofing

Deroofing, first described by Mullins et al. in 1959, is a simple, conservative, low-cost surgery for Hurley stage II and III lesions [19]. The procedure involves stripping the “roof” from all the abscess or sinus tracts and exposing the floor of the lesions in the affected areas. A metal probe or scissors gently probe to identify all communicating tracts, which are then deroofed. The gelatinous proliferating sinus tissue is then removed using a curette, scalpel, or moistened gauze [18,20,21]. The preserved floor allows the epithelial cells from sweat glands and hair follicle remnants to quickly re-epithelialize the wound and heal by secondary intention [22,23]. 

Deroofing is a simple and minimally invasive procedure that can be achieved by multiple methods, specifically, blunt surgical scissors, carbon dioxide (CO_2_) lasers, or electrosurgery probe. An open study by Vanderzee et al. treated 44 patients with a total of 88 lesions with a deroofing procedure followed by healing with secondary intention. Most deroofed lesions were located in the groin (47%) and axillae (44%). The mean healing time was 14 days in defects with a mean length of 3 cm. It was observed that 17% of the deroofed lesions recurred after a median of 4.6 months, while 83% of the lesions showed no recurrence after a median follow-up of 34 months. A total of 90% of the treated patients were willing to recommend the deroofing technique to others with HS [24].

Electrosurgery, with a mean healing time of around 16 days, also proved to be a good alternative for deroofing early HS lesions [25]. 

Deroofing presents many advantages: it can be performed under local anesthesia, offers low morbidity and cosmetically acceptable results, and prevents contractures [25]. Deroofing is the primary procedure for persistent nodules and sinus tracts in Hurley stage I or II. Complications associated with the procedure include post-surgical bleeding, infection, and scarring [26,27]. 

### 3.3. Excision

Excision is a more invasive approach that is aimed at removing the diseased tissue in its entirety. Depending on the level of invasiveness, excision can be limited, where each diseased area is excised separately, with a rim of normal tissue margin, or wide, where an area comprising all lesions is excised. In especially severe cases, radical excision of the entire area of a body region with disease activity may be indicated [28]. However, excision surgery is not uniformly defined in most studies, making the comparison of treatment outcomes challenging. Different studies may have described the same procedure under different names. This also leads to challenges in defining recurrence.

#### 3.3.1. Limited/Localized Excision

This office-based procedure can be performed under local anethesia. In their retrospective study on 57 patients with an average of 92 operated lesions, Van Rappard et al. described local excision as the complete resection of the diseased tissue beyond the borders of activity, leaving behind clear margins. The selection criteria for local excision included (i) recurrent abscesses or fistulas in the same location; (ii) Hurley stage I and II lesions where excision could be performed easily, leaving behind healthy tissue around and below the lesion; and (iii) their lesions were smaller than palm size, in order to not exceed the maximum quantity of lidocaine that could be used per procedure. The defect was managed with a primary closure, and patients were followed up for an average of 27 months. In total, 63% of patients were treated successfully, without any recurrence, 83% were satisfied with the cosmetic outcomes, and 89% were willing to suggest it to other patients with HS [29]. 

#### 3.3.2. Wide Excision

Wide excision should be offered in severe cases that are refractory to medical treatment when there is a risk of extensive fibrosis or architectural loss. Wide excision involves the surgical removal of lesions as well as the surrounding disease-free tissue, such as subcutaneous fat or a 1–2 cm lateral margin of intertriginous skin [15]. According to Park and Park’s recommendation, all apocrine glands should be removed from all hair-bearing skin to the deep underlying fascia [30]. Wide surgical excisions were defined by Alharbi et al. as the removal of diseased tissue with a wide margin of 1 cm up to the subcutaneous tissue until reaching the fascia. While some studies have a clear definition for wide excision, many studies do not mention a definite criterion, which makes it difficult to compare various studies [31]. Sinus tract location, extent, and fluid collections can be demonstrated preoperatively by magnetic resonance imaging or ultrasound [32]. They can also be demonstrated intraoperatively by dye mapping techniques, such as methyl violet and iodine starch [33,34], so that the entire area of HS lesions can be identified and removed. This is of particular importance in intertriginous areas, as they require the complete removal of diseased tissue by wide excision for a successful outcome [34]. Lelonek, et al. have reported their observations in a case series of seven patients with genital elephantiasis due to HS. In these patients, wide excision of the affected genital parts followed by surgical reconstruction using combined surgical methods was performed. The procedure resulted in an improvement in the quality of life of the patients [35].

Cryotherapy can be used persurgically to assess the extent of the lesion with minimum discomfort and is simple to perform [36]. A CO_2_ laser can be used for debulking, which further improves hemostasis and the visualization of the operative field [15]. 

#### 3.3.3. Radical Excision

Yet again, there is no clear definition for radical excision. Some authors have termed it as the removal of the entire hair-bearing area in the affected area, with a clear margin of at least 1 cm [37]. Nesmith et al. have described radical excision as the removal of diseased tissue up to the deep fat and fascia and additionally removed superficial lymph nodes to eliminate microbacterial foci [38]. 

As per a meta-analysis, the estimated rate of HS recurrence is 22.0% following local excision and 27% following deroofing, compared to a recurrence of 13.0% following wide excision [39]. With the axilla being one of the most commonly affected sites, there is a risk of injury to the brachial plexus and the axillary artery and its branches in severe cases requiring a radical excision [15]. Special care must be taken in cases of inguinal and perianal HS, as there are chances to damage the anal sphincter and the vaginal wall. Hypergranulation is the most common complication in patients that are treated with wide excision and left to heal by secondary intention [15].

## 4. Closure Techniques

Secondary intention healing is optimal to minimize the risk of recurrence following HS excision, though it may be challenging in cases with large defects, and other techniques can be considered. The closure technique is selected on the basis of various factors, including site, size and location of the defect, patient preference, risk of bleeding, graft necrosis, wound dehiscence, and preservation of function/range of motion. In the reconstruction of large defects, as in radical wide excisions, extra consideration is required to preserve function and provide acceptable aesthetic outcomes. Various options are available for closing the wounds, such as sutures, grafts, and flaps. In cases of high wound tension where wound edges cannot be approximated, wounds are left open to heal by secondary intention. As per a systematic review by Mehdizadeh et al. [39], the recurrence rates after wide excision are 15% with primary closure, 8% in flaps, and 6% in grafting. The study also mentions that healing with secondary intention had much lower recurrence rates; however, the rate was not mentioned.

## 5. Secondary Intention 

Healing by secondary intention is the process by which a wound, left open intentionally rather than reapproximated, fills in with granulation tissue and eventually re-epithelializes over time. It is a well-established option in wound management, but the healing process is prolonged in this method [40], and there is a high risk of scar formation [20]. Moist wound dressings (e.g., silastic foam dressing) should be applied to hasten healing. Healing by secondary intention is often used after wide local excision, especially in the more severe Hurley stages II or III, and has been demonstrated to have satisfactory functional and aesthetic outcomes [41,42]. In a study by Decker et al., 253 procedures of wide excision with healing by secondary intention were carried out in 84 patients; during a mean follow-up of 36.2 months, 37.6% of the procedures developed recurrence. Total remission of disease activity was achieved in 49% of patients, while in 13% the authors observed disease progression. Overall, patient satisfaction was high, and patients expressed that they would recommend the procedure to others [43]. However, in other studies, recurrence rates as low as 16.7% were observed with healing by secondary intention after a mean follow-up of 52 months [44]. In another retrospective case series, local recurrence was noted in 2 of 23 treated patients (8.69%) over a follow-up period of 1.02 years [41]. Overall, secondary intention has the lowest recurrence rates and may be the choice of surgery wherever possible [39].

## 6. Skin Grafts

Skin grafting may be indicated when primary closure or skin flaps are not feasible (e.g., in large wounds on the buttocks or thighs), when a shorter time to wound closure is an important concern, or when there is less effective wound contraction or slow wound healing via epithelization [15,45]. Split-thickness skin grafts (STSG), full-thickness skin grafts, and recycled skin grafts have demonstrated acceptable functional and aesthetic outcomes [15,46]. 

STSG has been successful in studies. Bohn et al. reported 138 cases of HS wherein 122 cases were treated with wide excision followed by STSG. Recurrence was noted in 33% of cases and 83% of the surveyed patients would recommend the procedure to other patients [47]. The healing time in STSG was faster when compared to healing by secondary intention [48]. Acceptable hemostasis by the graft allows for graft placement in a single procedure as opposed to the conventional procedure of graft placement on the granulating wound [12]. Despite a shorter duration to wound closure, patients preferred healing by secondary intention, owing to the absence of a painful donor area, greater freedom of arm or limb mobility, and relatively lower pain [49]. Vacuum-assisted wound closure in wide excision with split thickness grafting provided superior outcomes [50].

STSG is preferred over the full-thickness skin grafting due to the ease of harvest and less complicated transfer; however, its disadvantages include the need for prolonged immobilization of the arm, sequelae at donor sites, and, rarely, the formation of retractile scars [51]. Skin grafting after excision is associated with increased pain, immobilization, prolonged hospitalization, and longer healing times compared to skin flaps [52]. The absence of hair follicles and sweat glands in STSGs may be advantageous in HS because both hair follicles and sweat glands are involved in the pathogenesis of HS [32,45]. Most studies on skin grafts are limited to retrospective analyses. 

## 7. Skin Flaps 

Skin flaps are similar to skin grafts, but the flaps maintain an intact blood supply, whereas grafts depend on the growth of new blood vessels [20,26]. The advantages of skin flaps are that they provide thick tissue coverage and have shorter healing times, which can be especially important when functional disability is a concern in the postoperative period [53]. However, their use is limited due to a poor vascular supply to distant portions of the flap, leading to a high risk of ischemia and necrosis, and because they frequently require debulking due to their thickness [15]. Skin flaps are recommended when vascular channels and nerves are exposed [15]. Advances have been made in flap construction, and several types of flaps exist that can be employed in multiple areas of the body [15,41]. Flaps of particular importance in the reconstruction and closure of HS wounds are the lateral thoracic flap [53], fasciocutaneous V-Y flap, Limberg flap [54], and musculocutaneous flap [55,56]. Additionally, perforator flaps, such as the thoracodorsal artery perforator (TDAP) flap [57], have been reported as advantageous in regards to range of motion, site proximity, and skin quality. Similar to skin grafts, most literature is from case reports; therefore, further larger studies are required. 

## 8. Primary Closure

Primary closure is the use of sutures or staples for closure and is most often used to close smaller excisions, especially in lower-grade HS cases. However, if HS lesions are not effectively excised, disease can recur at the periphery and result in wound dehiscence [58]. Primary closure is associated with the highest recurrence rates among closure techniques [41]. In 100 cases treated with primary closure, Mandal et al. reported recurrence in 69.88% of cases [59]. However, Rappard et al. noted a 34% overall recurrence; they also noted that primary closure seemed more desirable and well-tolerated amongst patients, with 84% of the patients willing to undergo a repeat procedure if necessary and 89% willing to recommend the procedure to other patients [29]. Primary closures may be complicated by seromas, which require drainage if they are large. Other late complications include wound dehiscence and graft/flap necrosis, which may require debridement. Long-term complications include scarring and contractures at the repair site [15].

Vacuum-assisted closure with delayed primary closure was reported to be effective in wider wounds and showed reduced healing times when compared to primary closure alone [60].

## 9. Combination Reconstructions

In this procedure, multiple closure techniques are performed simultenously, employing the advantages of each technique to optimize the outcome [15]. In the example of the star-like technique, five equilateral triangles bordering foci of axillary disease are excised in addition to the central foci. The edges of each triangle are then sutured together to create a final considerably smaller scar. This method allows the wound to be partially sutured while leaving the remaining area to heal by secondary intention [61]. A small number of case series and prospective studies on combined reconstructions are available, but RCTs are lacking. 

Wide excision combined with continued aggressive medical management and dietary modifications is effective in providing functional long-term results, as wide excision is associated with a low recurrence rate [6]. However, it not only predisposes patients to larger wounds, surgery-site infection, and prolonged recovery periods compared to local excision or I and D [32], but it also carries the risk of injury to the neurovasculature and other vital structures, contributing to greater postoperative morbidity [62]. However, in one study, 204 of 255 (80%) patients were markedly satisfied with the postoperative outcomes of wide excision [34], supporting the idea that, although the potential complications may be more severe, the long-term improvements in quality of life make wide surgical excision a first-line treatment option in all stages of HS [34,38]. Alternatively, when effective wound healing and cosmetic results are of utmost importance, wide excision may be followed by other less invasive procedures, such as vacuum-assisted closure (VAC therapy), platelet-enriched plasma, and dermal substitutes [15], though these lack RCTs.

## 10. Skin-Tissue-Sparing Excision with Electrosurgical Peeling

The skin-sparing excision with electrosurgical peeling was introduced by Blok et al. [26] in 2015. It is an alternative to wide surgical excision for Hurley stage II or III HS and is associated with many advantages over other procedures. With this technique, healing time is reduced, complication rates are low, and cosmetic outcomes are much better compared to other procedures. In this procedure, an electrosurgical wire loop is passed successively in tangential transections over the sinus tract until the epithelialized floor of the sinus tract is exposed. The margins are inspected and probed for the presence and subsequent removal of any residual sinus tracts. The wound should be allowed to heal by secondary intention, and steroids should be injected to prevent hyper-granulation. The method for this procedure is almost the same as in the deroofing method, wherein all lesional tissue is removed while leaving the floor of the tract intact, which leads to rapid wound healing, a low risk of contracture, and excellent cosmetic outcomes [26,63]. Currently this procedure lacks RCTs with long-term follow-ups. 

## 11. Lasers and Intense Pulse Light

### 11.1. CO_2_ Laser

A CO_2_ laser was first used in the treatment of HS by Dalrymple and colleagues in 1987 [64]. Three different techniques can be used to excise tissue with a CO_2_ laser: CO_2_ excision, CO_2_ laser stripping, and CO_2_ laser vaporization 

The diseased tissue, comprised of boils, sinus tracts, granulomatous tissue, and scars, are vaporized by a CO_2_ laser, then closed, either by primary suturing or left to heal by secondary intention. The immediate hemostasis provides a bloodless field with excellent visibility to identify all sinus tracts while maximizing the preservation of the surrounding normal tissue [65,66]. Vaporization of the tissue by a CO_2_ laser under local or general anesthesia by repeated laser passes is useful in exploring down to the subcutis, unmasking the proliferating sinus tissues [67]. It is ideal for recurrent extensive lesions that were treated previously and are associated with fibrosis and scar formation. A recurrence rate of 29% was noted in one study [68]. 

In a study by Lapin et al., 4 (11.7%) of 34 patients were treated with CO_2_ laser stripping, where diseased tissue was vaporized in a stepwise horizontal manner and monitored preoperatively for any sinus remnants. This method was analogous to ‘macro Moh’s surgery’, and the mean healing time was 4 weeks. Surgical site recurrence was noted in two patients after a mean follow-up period of 34.5 months. However, 12 cases (35.3%) had progression of disease activity in the surrounding area >5 cm from the site of excision [69]. Remission has been reported to last up to 12 months or more [70]. 

A study of CO_2_ laser excision with marsupialization followed by healing by secondary intent using a high-energy focused mode demonstrated a low recurrence rate; after the treatment of 185 areas in 61 HS patients, only 2 patients reported recurrence [71].

The advantages of CO_2_ laser treatment include improved healing and preservation of normal tissue and the reduced disfigurement of treatment sites in comparison to surgical excision with or without grafting. Additionally, postoperative pain and discomfort were more tolerable with a CO_2_ laser [72]. Post-operative granulation tissue formation, infection, and cellulitis were the most common complications associated with the CO_2_ laser modalities [73].

### 11.2. Nd: YAG Laser

Long-pulsed Nd:YAG (1064 nm) is a novel treatment option that acts through selective photothermolysis of the follicular unit, thus reducing pain, inflammation, and suppuration. It also reduces the frequency of HS recurrence. For Fitzpatrick skin types I–III, the recommended starting settings are fluence 40–50 J/cm^2^, pulse duration 20 ms, and spot size 10 mm; for Fitzpatrick skin types IV–VI: fluence 35–50 J/cm^2^, pulse duration 35 ms, and spot size 10 mm [72].

Prospective randomized studies have shown a 72% mean reduction in HS activity in all affected regions after four monthly sessions of Nd-YAG laser therapy [74]. Not only does it treat the existing HS lesions, but it also offers a preventative effect against the recurrence and emergence of new boils, abscesses, and sinuses. Authors have also noticed a reduction in the requirement of systemic drug therapy both during and after the completion of laser therapy. 

The response rates with Nd-YAG laser treatment are best observed in areas with dark coarse terminal hair, such as the axilla and the inguinal areas, whereas the buttocks and inframammary areas are relatively resistant to treatment, likely due to the presence of short, less pigmented vellus hairs in these areas. This also reiterates the role of dysregulation of the follicular milieu in the pathogenesis of HS [75]. Histopathological evaluation of lesions treated with Nd-YAG have revealed follicular-oriented lymphocytic infiltrates post-treatment, which confirms the follicular origin of the disease and the role of Nd-YAG lasers in curbing disease progression [74,76].

### 11.3. Intense Pulsed Light (IPL)

IPL successfully reduced disease severity compared to the control side in a randomized controlled trial in 18 patients with HS. These patients were randomized to receive IPL twice a week for 4 weeks. The results were maintained at 12 months and had good patient satisfaction [77]. 

### 11.4. Others

Laser hair removal with other modalities, such as the long-pulsed alexandrite laser (755 nm), has achieved favorable outcomes in three case reports, with a significant improvement in symptoms and no recurrence in follow-up periods ranging from 10 months to a year [78,79,80].

A diode laser was also reported to achieve a partial to favorable improvement of hidradenitis suppurativa in small case reports [81,82]. 

## 12. Cryoinsufflation

Cryoinsufflation is a novel method that was first described in 2014 for the treatment of Hurley stage II–III HS [83]. It is a form of modified spray cryotherapy wherein liquid nitrogen is directly injected into the sinus tract. Once the lesion is identified and local anesthesia is administered, a 21-gauge needle attached to a cryosurgical unit is inserted into the opening of a sinus tract, and liquid nitrogen is sprayed into the tract for a typical pulse duration of 5 s followed by a 2–3 s pause; the same process is repeated three times for each lesion. The same treatment protocol is followed at monthly intervals until the tract is obliterated. Satisfactory results were initially obtained in two patients treated in this manner. Later, two modifications were added: the first is the prescription of systemic antibiotics prior to the procedure, and the other is the use of a 21-gauge olive-tipped cannula to decrease the risk of adverse events, such as air embolism [84]. The effectiveness and safety of drainage and cryopunch i.e., punch-trocar-assisted cryoinsufflation (cryopunch), has also been evaluated in inflammatory acute fluid collection [85].

## 13. General Care, Post-Surgical Care, and Recovery

In cases where hemorrhage occurs, it can be controlled by either the application of pressure, in cases of small vessel injury, or the ligation of large vessels to control bleeding. Infection of the wound in the early post-procedure period may be avoided by meticulous cleansing and dressing of the wound and prescribing topical and oral antibiotics [15]. 

In procedures, such as deroofing, STEEP, and wide local excision, where the wound is left to heal by secondary intention, more rapid healing is facilitated by daily wound cleansing and moist dressings. Alginate or silicone dressings help facilitate quicker and improved wound healing [24,63]. In the later weeks, thick absorbent dressings made of gauze soaked in an equal mixture of petroleum and liquid paraffin are placed following irrigation with disinfecting solutions. These gauze dressings are changed every 2 to 3 days. Vacuum-assisted negative pressure dressings can be used in grafts to increase the local oxygen concentration in the wound, reduce the bacterial growth, improve healing, and keep the graft intact on curved locations [86,87]. 

Pain management in HS begins with disease control but can be supplemented with short-acting opioid analgesics in acute cases and following surgical procedures. Chronic pain can be managed according to the World Health Organization pain ladder as per recommendations of the North American clinical management guidelines for hidradenitis suppurativa [9].

Depending on the site and size of the defect, it can take 4 to 10 weeks to achieve a full recovery with a preoperative range of motion [15]. Physiotherapy should be initiated early in the course of the recovery to prevent wound contractures [88]. Lifestyle changes, such as the cessation of smoking, weight loss, and the treatment of other co-morbidities, should be initiated in the recovery phase to prevent recurrence [6,9,10].

## 14. Recurrence

Recurrence is not uniformly defined in most studies. One study defined recurrence as the reappearance of diseased tissue within 0.5 cm of the surgical site [24], and another study defined recurrence as “an inflammatory boil immediately within the scar or within less than 0.5 cm from the scar [87].” However, most authors have defined recurrence as the reappearance of inflammatory tissue at the operated site or based on the site if it is involved in recurrence [44]. Few studies have measured recurrence by the number of new lesions appearing after surgery, and few have measured recurrence according to the region involved with new inflammatory nodules [89]. This heterogeneity in definition and study protocols make the comparison of different modalities difficult. Incision and drainage had a recurrence rate of 100% and, thus, is not preferred as a primary modality of surgical management. A meta-analysis by Mehdizadeh et al. reported that the highest recurrence rate is seen following deroofing, (27.0%) followed by local excision (22.0%). Wide excisions were preferable due to a low recurrence rate of 5–13.0% [41,44]. Within the wide excision group, recurrence rates by closure were 15% with primary closure, 8% for flap closure, and 6% for grafting. Secondary intention healing had a much lower average recurrence and is, thus, preferred by most [41]. At least 30% of patients who underwent any form of surgery developed a recurrence requiring a second revision surgery. Of those who underwent a second surgery, 25% had a relapse; 23% of those who underwent a third surgery developed recurrence. The morbidity associated with repeated surgeries limits the preference of surgical methods over more conservative medical treatment [90]. 

Recurrence is principally due to the inadequate resection of sweat glands from the site and is also influenced by other factors such as obesity, smoking, local friction, and maceration. HS, being an inflammatory disease, progresses to previously unaffected areas in about 20–25% of patients [20,37,68]. Younger age and multiple sites were associated with higher recurrence, and the site of surgery often was not a factor affecting recurrence in long term follow-up [16].

## 15. Peri-Surgical Use of Biologics for the Treatment of HS

Adalimumab was found to be safe and efficacious in the treatment of HS peri-surgically (SHARPS study) without the need for drug interruption prior to the surgery. At week 12, a significant number of patients in the adalimumab group achieved favorable outcomes across all bodily regions compared to placebo, and it was not associated with grave side effects [91]. Similar results were found with the use if infliximab (IFX) prior to surgical therapy, and the results were superior compared to IFX alone as assessed by HS PGA and aided in long-term clearance [92]. Another study investigated biologic therapy with either IFX or ustekinumab following the surgical resection of HS lesions. Lower recurrence rates were experienced (19%) in patients treated with infliximab compared with placebo (38.5%) [93].

## 16. Choice of Surgical Treatment and Recommendations

The process of decision making while opting for a surgical treatment depends on the chronicity of the disease, the affected area, the extent of inflammation, the patient’s comorbidities, prior surgery and scars, and, most importantly, a patient’s opinion and willingness to undergo surgery. In acute cases, simple procedures, such as local incision and drainage, unroofing, and marsupialization, may be performed for quick relief, but recurrence is often inevitable. After the regression of acute disease, limited local excision or wide excision of the affected site can be performed. Mapping techniques (e.g., iodine starch and methylene blue) are crucial for the complete removal of lesions during excision [94]. The preferred method of closure depends on the site and extent of excision and also on the experience of the treating surgeon. There is, however, no consensus on the type of wound closure for a particular wound size. In retrospective studies, the wound sizes at operation ranged from 30 cm^2^ to over 1500 cm^2^.

A total of 83 wounds with a median size of 159 cm^2^ were left open to heal completely with secondary intention at the time of surgery, while 117 wounds with a median size of 100 cm^2^ were partially closed. Smaller wounds with a median size of 38 cm^2^ (30 wounds) were closed primarily, and 15 wounds with a median size 196 cm^2^ were closed by skin graft. Smaller wounds benefited from primary/partial closure. However, secondary intention healing is the goal wherever possible. In large wounds, healing can be facilitated by delayed grafting and/or flap reconstruction [46]. However, they are not without the risk of failure, depending on the surgeon’s experience.

### Guideline Recommendations

There are several regional and international guidelines for the treatment of HS. However, the dearth of controlled studies has made it difficult to reach a consensus as to when surgery should be considered and the type of surgery to be considered. The existing data are comprised of case series, cohorts, and retrospective studies with no clearly defined outcome measures. 

The most recent international HS ALLIANCE Guidelines recommend surgical options to be considered in severe recurrent cases of HS that have not been controlled by systemic medications, which is consistent with the recommendations by the European S1 guidelines for the management of HS. They recommend the drainage of tense abscesses in acute scenarios for immediate relief, but this should not be used as the sole mode of treatment and must be combined with topical or systemic therapy for optimal results and prevention of recurrence [6,11].

The North American clinical management guidelines for HS also recommended incision and drainage only in acute tense abscesses. For recurrent sinuses and nodules, they recommend deroofing, STEEP, or electrosurgical excision. Local wide excision can be performed for extensive severe disease; however, there is no consensus on closure technique. The guidelines also recommend Nd-YAG laser for Hurley stage II or III disease. A CO_2_ laser can be used in Hurley stage II or III disease with fibrotic scarring and nodules. Perioperative medical treatment has been beneficial in achieving better outcomes [9]. 

The British Association of Dermatologists guidelines for the management of HS recommended surgical treatment at the very end of their algorithm in recurrent and severe cases not responding to oral antibiotics, acitretin, or adalimumab [10].

Although the combined medico-surgical approach to HS has insufficient evidence, its importance has been strongly recommended by the Centre of Evidence of the French Society of Dermatology in their recommended guidelines for the treatment of HS [95]. 

Table 1 summarizes surgical procedures in HS with the advantages and disadvantages of each method.

## 17. Conclusions

HS is a chronic debilitating inflammatory disease of the skin with a great impact on patients’ physical, social, and functional quality of life. Therefore, it requires effective and timely treatment. Treatment of HS can be accomplished by both medical and surgical modalities, depending on its clinical stage and severity, typically requiring a combined approach, highlighting the need for a multidisciplinary treatment protocol. Each surgical method has its own advantages and limitations. Considering the advantages and limitations of each method, clinicians should select the best possible surgical method for an individual patient. The choice of procedure should be individualized. Discussion about different surgical methods will help to improve the awareness of treatment options in HS and enable clinicians to confidently select the best management for each individual patient.

## Figures and Tables

**Table 1 jcm-11-02311-t001:** Summary of Surgical Procedures in Hidradenitis Suppurativa.

Ref.	Procedure	Hurley Stage	Level of Evidence/Strength of Recommendation (7)	Mean Healing Time	Recurrence Rate	Pros	Cons	Complications
[17,18]	**Incision and drainage**	Ⅰ/Ⅱ	Ⅱ/C	7 to 10 days	≈100% (12, 13)	Immediate release of tension Immediate pain relief in acute abcesses Minimally invasive (10)	Reccurence Temporary treatment	Infections Sinus formation
[19,20,21,22,23,24,25,26,27]	**Deroofing** **(cold steel incision, CO_2_ laser, and electrosurgery)**	Ⅰ/Ⅱ	Ⅱ/B	14 days	27% (34)	Minimally invasive Preserves normal sorrounding tissue Immediate pain relief	Temporary. Does not remove diseased tissue entirely	Infection Hypergranulation tissue in the wound bed (11)
[41,42,43,44]	**Wide excision with healing with secondary intention**	Ⅱ/Ⅲ	Ⅱ/C	6 to 12 weeks (12)	37.6% (38)	Better cosmetic healing No need for donor tissue Acceptable limb mobility	Longer healing time Regular dressing changes and wound care	Postoperative bleeding Exposure of vessels and nerve plexus Secondary infections Contractures
[29,41,58,59,60]	**Wide excision with primary closure**	Ⅱ/Ⅲ	Ⅱ/C	3.2 weeks (41)	34% to 66% (40, 41)	Simple Faster healing Less contracture Better patient satisfaction	Suitable for small wounds Wound dehiscence	Suture dehiscence Infections Seroma Keloid
[45,46,47,48,49,50]	**Wide excision with skin grafts**	Ⅱ/Ⅲ	Ⅱ/C	6 weeks (44)	33% (45)	Reduced healing time Cosmetically and fuctionally better	Graft dehiscence and necrosis Donor site infection, pain, and scarring Reduced mobility	Graft necrosis Infection Graft contracture Seroma Cellulitis
[26,53,54,55,56,57]	**Wide excision with flaps**	Ⅱ/Ⅲ	Ⅱ/C	2–4 weeks (70)	19%	Best method for skin closure Avoids bad scarring	Difficulty in harvesting. Requires expertise Vascular insufficiency and necrosis Postoperative pain and morbidity	Brachial plexus damage Flap necrosis Wound dehiscence Haemorrhage
[63]	**STEEP**	Ⅱ/Ⅲ	Ⅳ/D	Not known	Not known	Tissue sparing effect Heals faster Good hemostasis	Recurrence Low evidence	Scar formation and contracture Infections
[64,65,66,67,68,69,70,71,72,73]	**CO_2_ laser** **excision**	Ⅰ/Ⅱ	Ⅱ/C	8 to 10 weeks (63)	18% (63)	Immediate hemostasis Bloodless field that offers clear view of surgical site Tissue sparing property	Scar formation Chance of recurrence	Postoperative pain Scarring Functional restriction Cellulitis
[74,75,76]	**Nd-YAG laser (1064 nm)**	Ⅰ/Ⅱ	Ⅱ/B	1–2 weeks	Not known	Minimally invasive Reduce the follicle count and thereby eliminate the cause Less scarring Rapid healing	Postoperative pain Recurrence Limited efficacy in long-standing disease	temporary paresthesias

Strength of Recommendation Taxonomy recommendation level: I, good-quality patient-oriented evidence; II, limited-quality patient-oriented evidence; and III, other evidence, including consensus guidelines, opinion, case studies, or disease-oriented evidence. Evidence grading level: A, recommendation based on consistent and good-quality patient-oriented evidence; B, recommendation based on inconsistent or limited-quality patient-oriented evidence; and C, recommendation based on consensus, opinion, case studies, or disease-oriented evidence [96].

## Data Availability

Not applicable.

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
