# Peer review of "Surgical Treatment in Hidradenitis Suppurativa"

_jcm, 2022, doi:10.3390/jcm11092311_

Round 1

Reviewer 1 Report

This is an interesting narrative review about surgical treatment in hidradenitis suppurative. Some recommendations:

In the abstract you mention that you included case reports, case series and RCTs. I suppose you also included cohort and case-control studies. Please add

I would add the prevalence of this disease in the introduction.

Citations should be added to the impairment of quality of life of this disease, for example (10.3390/ijerph18136709, 10.3390/ijerph18116131)

The role of ultrasound in the surgical management of hs should be added (10.1111/jdv.16435)

It would be interesting that you add a table comparing the types of closure and the (recurrences, complications…)

I think you could include all laser in the same section and omit the title of hair removal

Regarding Cryoinsufflation, you should also include information about the cryopunch (10.1111/jdv.154069)

I think the complications, cares and recurrences of each technique should be included in each section and not as a final paragraph. Maybe you could add a final paragraph with general care.

I think you should split some parts of the “recommendation” section and add a new section about the management of medical and biologics treatments peri-surgically.

You could add your opinion about what is the best option for the surgical treatment of HS in the conclusion

Author Response

Date: 8th March 2022

To,

The Editor in Chief

JCM

Sub: Revised manuscript “Surgical Treatment in Hidradenitis Suppurativa”

Dear Sir,

Thank you for your email. We really appreciate valuable comments of the reviewers and opportunity to improve the manuscript. We have revised the manuscript based on the comments and suggestions from the esteemed reviewers. Please find below point to point clarifications to the comments. Revised manuscript with changes highlighted in red colour is attached in the email along with this letter. We request you please consider revised manuscript for publication.

Reviewer 1

This is an interesting narrative review about surgical treatment in hidradenitis suppurative. Some recommendations:

  1. In the abstract you mention that you included case reports, case series and RCTs. I suppose you also included cohort and case-control studies. Please add

Response: We have added cohort studies and case-control studies. It is included in the abstract

  1. I would add the prevalence of this disease in the introduction.

Response: Prevalence of the disease is added in the introduction.

  1. Citations should be added to the impairment of quality of life of this disease, for example (10.3390/ijerph18136709, 10.3390/ijerph18116131)

Response: Citations are added for the impairment of quality of life.

  1. The role of ultrasound in the surgical management of hs should be added (10.1111/jdv.16435)

Response: The role of ultrasound in the surgical management of HS is added.

  1. It would be interesting that you add a table comparing the types of closure and the (recurrences, complications…)

Response: Thank you for your valuable suggestion. However, as all this information is not available from all the studies. Hence, separate table is not prepared.

  1. I think you could include all laser in the same section and omit the title of hair removal

Response: Role of laser in clubbed in one section and title of hair removal is deleted.

  1. Regarding Cryoinsufflation, you should also include information about the cryopunch (10.1111/jdv.154069)

Response: Information about the cryopunch is added.

  1. I think the complications, cares and recurrences of each technique should be included in each section and not as a final paragraph. Maybe you could add a final paragraph with general care.

Response: Changes done as suggested.

  1. I think you should split some parts of the “recommendation” section and add a new section about the management of medical and biologics treatments peri-surgically.

Response: Changes done as suggested.

  1. You could add your opinion about what is the best option for the surgical treatment of HS in the conclusion

Response: Considering the advantages and limitations of each method, clinicians should select best possible surgical method for an individual patient. Choice of procedure should be individualized considering patient’s characteristics and severity of condition. It is difficult to suggest best option. We have added this in the conclusion section.

Best regards,

Dr. Mohamad Goldust

Reviewer 2 Report

The present review entitled “surgical treatment in hidradenitis suppurative (HS)” focuses on various surgical options available for the treatment of HS. The current review covers all available information regarding surgical options used in the treatment of HS and summarized it in a very narrative and informative manner. Although, HS treatment can be accomplished by both medical and surgical modalities depending on its clinical stage and severity. However, sometimes it requires a combined approach, highlighting the need for a multidisciplinary treatment protocol. The objective of this review is to increase the awareness of treatment options in HS and enable clinicians to confidently select the best management for each individual patient. This review has been executed with care, and can be used by scientists since it provides adequate information specific for clinician as well as basic researchers, who wish to study options for HS treatment. 

Author Response

Date: 8th March 2022

To,

The Editor in Chief

JCM

Sub: Revised manuscript “Surgical Treatment in Hidradenitis Suppurativa”

Dear Sir,

Thank you for your email. We really appreciate valuable comments of the reviewers and opportunity to improve the manuscript. We have revised the manuscript based on the comments and suggestions from the esteemed reviewers. Please find below point to point clarifications to the comments. Revised manuscript with changes highlighted in red colour is attached in the email along with this letter. We request you please consider revised manuscript for publication.

Reviewer 2

  1. The present review entitled “surgical treatment in hidradenitis suppurative (HS)” focuses on various surgical options available for the treatment of HS. The current review covers all available information regarding surgical options used in the treatment of HS and summarized it in a very narrative and informative manner. Although, HS treatment can be accomplished by both medical and surgical modalities depending on its clinical stage and severity. However, sometimes it requires a combined approach, highlighting the need for a multidisciplinary treatment protocol. The objective of this review is to increase the awareness of treatment options in HS and enable clinicians to confidently select the best management for each individual patient. This review has been executed with care, and can be used by scientists since it provides adequate information specific for clinician as well as basic researchers, who wish to study options for HS treatment.

     Response: Thank you for your positive comments. There are no suggestion from reviewer 2.

Best regards,

Dr. Mohamad Goldust